# EXAMINING WHY PERTURBATION-BASED FIDELITY METRICS ARE INCONSISTENT

## ABSTRACT

Saliency maps are commonly employed as a post-hoc method to explain the decision-making processes of Deep Learning models. Despite their widespread use, ensuring the fidelity of saliency maps is challenging due to the absence of ground truth. Researchers, therefore, have developed fidelity metrics to evaluate the fidelity of saliency maps. However, prior investigations have uncovered statistical inconsistencies in existing fidelity metrics using multiple perturbation techniques without delving into the underlying causes. Our study aims to explore the origins of these observed inconsistencies by examining the existing fidelity metrics and demonstrating why they are inconsistent. We use different types of perturbations and study multiple models across different datasets. We propose two conformity measures to examine the validity of the assumptions made by the existing fidelity metrics. Our findings reveal that the assumptions made by the existing fidelity metrics do not always hold, making them inconsistent and unreliable. Thus, we recommend a cautious interpretation of fidelity metrics and the choice of perturbation technique when evaluating the fidelity of saliency maps in eXplainable Artificial Intelligence (XAI) applications.

## 1 INTRODUCTION

Deep learning (DL) models, while providing high performance and accuracy for various applications, come at the cost of decreased transparency. In many critical domains such as health care, insurance, and law enforcement, concerns about the transparency, fairness, privacy, and trustworthiness of AI applications arise due to the black-box nature of deep learning models (Rudin, 2019; Jacovi et al., 2021; Arrieta et al., 2020). These concerns have led to discussions about adopting the latest Artificial Intelligence (AI) models in various sectors (Cubric, 2020; Cam et al., 2019; Güngör, 2020). Therefore, a great deal of research has been dedicated to explaining the decisions of AI systems under the umbrella of XAI (Arrieta et al., 2020; Selvaraju et al., 2017; Chattopadhay et al., 2018; Zhou et al., 2016; Ramaswamy et al., 2020; Ribeiro et al., 2016; Broniatowski et al., 2021; Lundberg & Lee, 2017).

Saliency maps (e.g., Class Activation Maps (CAM)) are widely used as a mode to explain the decision of DL models (Selvaraju et al., 2017; Chattopadhay et al., 2018). Disagreements can be observed among saliency maps generated using different methods for the same model and the same image, making a user choice difficult. One can choose the best saliency map with the highest fidelity when compared to ground truth. However, the absence of actual ground-truth[1]. Fidelity metrics such as "Area Over the Perturbation Curve" ($AOPC$) (Samek et al., 2016), Average Drop (AD%), Increase in Confidence (IC%) and Win (W%) (Chattopadhay et al., 2018; Wang et al., 2020) and "faithfulness" metric (Alvarez Melis & Jaakkola, 2018) have been used to measure the fidelity of saliency maps (Samek et al., 2016; Bach et al., 2015; Alvarez Melis & Jaakkola, 2018).

These fidelity metrics, however, suffer from inconsistencies and thus make them unreliable (Tomsett et al., 2020). Fidelity metrics such as $AOPC$ (Samek et al., 2016), AD%, IC% and W% (Chattopadhay et al., 2018; Wang et al., 2020) and $faithfulness$ (Alvarez Melis & Jaakkola, 2018) rely on

---

[1]Human annotation typically focuses on features that make sense from a human perspective (e.g., edges in images), while DL models rely on patterns that are not easily interpretable. Human-annotated saliency maps may misrepresent the model's true decision-making process, making them unreliable for evaluating the fidelity of the maps.

computing pixel importance rank (PIR) for measuring the fidelity of saliency maps. PIR is calculated by perturbing the pixels (one by one or cumulatively) and noting the change in the output probability. A greater change in output probability denotes greater importance for a perturbed pixel. The computed PIR from an image serves as a proxy for ground truth, enabling the estimation of the fidelity score for saliency maps (Alvarez Melis & Jaakkola, 2018). This approach is based on the assumption that the change in output probability follows a consistent pattern across different perturbations, with the output probability varying in proportion to the importance of the perturbed pixel. If this assumption is not fulfilled, the fidelity metrics' scores would vary for different perturbations, leading to inconsistency as reported by Tomsett et al.(Tomsett et al., 2020). Further, Tomsett et al.(Tomsett et al., 2020) observed this inconsistency by analyzing the prediction probabilities by perturbing pixels with 0 and a random value. While demonstrating the inconsistency in fidelity metrics, Tomsett et al.(Tomsett et al., 2020) further recommend:

> "Metric developers should encourage users of their metric to investigate and understand the sources of variance in the metric scores, and how this affects their decisions about what saliency methods to choose for their particular model."

Thus, complementing the previous work by Tomsett et al. (Tomsett et al., 2020), we investigate the construction of fidelity metrics by studying the variances.

## 1.1 OUR CONTRIBUTIONS

We first theoretically establish the scenarios under which such assumptions are violated. We then provide two conformity measures that quantify the extent of variances affecting the fidelity metrics. Both the conformity measures are used to demonstrate the inconsistency of fidelity metrics by using several perturbations, models and datasets in both normal and adversarial setting. Going beyond the works of Tomsett et al.(Tomsett et al., 2020) and to generalize our findings, we study the variances in a comprehensive manner using nine different perturbations that include two inpainting-based perturbations (Telea (Telea, 2004) and Navier Strokes (Bertalmio et al., 2001)), Gaussian Blur (three different widths of the Gaussian Kernel) and setting a random value, min, max and mean of the image pixel values as perturbation values. Further, we show empirically that our conformity measures can be used in pixel-wise and segment-wise perturbation schemes before using fidelity metrics.

Our main contributions to this paper are:

- We present an approach to examine the inconsistency of fidelity metrics. We show that before using fidelity metrics, the varaiances of DL models w.r.t. to the perturbation type must be studied.
- Complementing previous works that have observed inconsistencies in fidelity metrics, we propose two new conformity measures.
- The conformity measures proposed in this work are further used to empirically analyse three widely used DL models and two adversarially trained DL models on three datasets using nine perturbation types, and two perturbation schemes (pixel-wise and segment-wise) for all models.

## 2 PROPOSED APPROACH

The fidelity metrics are based on the PIR which assume the drop in output prediction probability of a DL model to be proportional to the relevance of the perturbed pixel (i.e., more important the pixel, larger the drop in output probability). The pattern of change (i.e. the proportionate change in output probability as per the relevance of the perturbed pixel) should ideally hold true for all types of perturbations as long as the image semantics is preserved under the notion of local neighborhood. This is based on two aspects:

[P1] There is a drop in the output probability when a pixel is perturbed;

[P2] The amount of drop in output probability is proportional to the relevance of the pixel.

Dissecting these two aspects, we first present the theoretical background on the violation such aspects in fidelity metrics and then present the proposed conformity measures in Section 2.2 and Section 2.3 to aid in examining the inconsistencies.

## 2.1 THEORETICAL FRAMEWORK

Let $\mathfrak{R}$ be the ranks of pixel as per importance obtained from a saliency map on an unperuturbed image. $\mathfrak{R}$ can be expressed as follows:

$$\mathfrak{R} = \{a_1, a_2, a_3, a_4, \ldots a_i\} \tag{1}$$

where, $\mathfrak{R}$ is the ranked list of pixel importance by any saliency method. $a_1 \rightarrow a_i$ are pixels sorted in the order of their importance i.e. a greater $i$ denotes greater importance.

The assumption on the expected change in output probability by perturbing a pixel can be summarized as:

$$p_0 > p_i^\phi \quad \forall \quad i, \phi \tag{2}$$

where, $p$ is the prediction probability of a classification model which takes an image $I$ as input and returns the probability of the top class. $p_0$ is the probability of the top class as predicted for the original i.e. unperturbed image. $p_i^\phi$ is the prediction probability on an image obtained by perturbing only the $i^{th}$ pixel of an image $I$ with a perturbation type $\phi$.

Further, the change in output probabilities of perturbing two pixels $i$ and $j$, where $j$ is more important than $i$, can be summarized given as:

$$\delta p_i^\phi < \delta p_j^\phi \quad \forall \quad i < j \tag{3}$$

Where, $\delta p_i^\phi = p_0 - p_i^\phi$

Utilizing Equation (1) and Equation (3) we can generate the ranked list of probability differences, denoted as $\mathfrak{R}(\phi)$, for an image perturbed by each pixel and for all $i$ pixels with increasing order of ranks:

$$\mathfrak{R}(\phi) = \{\delta p_1^\phi, \delta p_2^\phi \ldots \delta p_i^\phi\} \tag{4}$$
$$pixels = \{1, 2, \ldots i\} \text{ and for a given perturbation } \phi$$

The probability changes obtained from Equation (4) can be sorted to get an ordered list of pixels. This set of ordered pixels, denoted by $R_\sigma$, represents the importance ranks of the pixels corresponding to $\sigma$. For a perturbation based technique to be applicable in fidelity metrics, the pixel importance ranks should ideally be invariant to different sets of hyper-parameters. This invariance to different sets of hyper-parameters is defined as below:

$$rbo(\mathfrak{R}(\phi), \mathfrak{R}(\psi)) \approx 1 \quad \forall \quad \text{for two perturbations} \quad \phi, \psi \tag{5}$$

Where, $rbo$ is Rank Biased Overlap (Webber et al., 2010) in our experiments, but it can be any function that calculates the similarity between two rank lists. Further, without the loss of generality we can say that Equation (5) should hold true for any set of pixels obtained from a saliency map.

Any perturbation based fidelity metric should conform to Point [P1] according to Equation (2) and should conform to Point [P2] according to Equation (5). To quantify the conformance, we introduce two new conformity scores which we refer to as $DROP$ (corresponds to Point [P1]) and $PSim$ (corresponds to Point [P2]) as discussed further.

## 2.2 DROP IN PREDICTION PROBABILITY (DROP)

The Drop in Prediction Probability ($DROP$) measures the average number of drops in the output probability when a pixel is perturbed for an image and a given model $M$. Thus, if $p_0$ represents prediction probability from a model $M$ on unperturbed image and $p_s^\phi$ represents the prediction probability on a perturbed image for a perturbation type $\phi$ on a chosen pixel $s$ in a set of all pixels $S$ or a chosen segment of all available segments, $DROP_\mathcal{M}$ for a given model can be computed as:

$$DROP_\mathcal{M} = \frac{\sum_{s \in \mathcal{S}} \left[ (p_0 - p_s^\phi) >= 0 \right]}{|\mathcal{S}|} \tag{6}$$

Where, [] denotes an indicator function with binary decision. For a complete dataset of $K$ images and a given model $M$, Equation (6) can be represented as Equation (7) providing average across all images in a dataset $D$.

$$DROP = \frac{1}{|\mathcal{K}|} \sum_{k=1}^{\mathcal{K}} DROP_\mathcal{M}^k \tag{7}$$

$DROP$ should have an ideal value of 1 but higher values i.e. closer to 1 are better under the assumption that there is a drop in the output probability when a pixel is perturbed.

## 2.3 PIXEL RANK SIMILARITY (PSIM)

For any two given set of perturbations (say $\phi$ and $\psi$) on an image, and corresponding ranked list obtained $\mathfrak{R}(\phi), \mathfrak{R}(\psi)$ respectively for a given image, it is expected to have same ranks for a given model $M$ if a model $M$ is consistent. Thus, the similarity between the ranks can be computed as:

$$PSim_\mathcal{M} = \frac{\sum_\phi \sum_{\psi, \phi \neq \psi} rbo(\mathfrak{R}(\phi), \mathfrak{R}(\psi))}{|\mathcal{N}| \times (|\mathcal{N}| - 1)} \tag{8}$$

Extending the same over the dataset $D$ with a set of $K$ images, $PSim$ can be computed as an average as Equation (9):

$$PSim = \frac{1}{|\mathcal{K}|} \sum_{k=1}^{K} PSim \tag{9}$$

Thus, for any perturbation based fidelity metric to be consistent, $PSim$ should have an ideal value of 1. However, higher values i.e., closer to 1 suggest the conformance of fidelity.

## 3 IMPLEMENTATION DETAILS

### 3.1 APPROACH OVERVIEW

Figure 1 shows our implementation where we obtain the prediction probabilities for a given model on unperturbed and a set of perturbed images. The prediction probabilities are used to evaluate the conformance using Drop in Prediction Probability (DROP) for Point Item [P1] and Pixel Rank Similarity (PSim) for Point Item [P2]. The approach for measuring the conformity scores is further described in Algorithm 1. While Algorithm 1 computes the conformity scores for the pixel-wise perturbation scheme, the same can be applied to the segment-wise perturbation scheme without the loss of generality.

We first determine the prediction probability of a given model $M$ on an unperturbed image (i.e., $p_0$) and then perturb the selected pixels one by one for a given perturbation $\phi_1$ to obtain $p_1, p_2, p_3 \ldots$ to determine the $\delta p_1, \delta p_2, \delta p_3 \ldots$ for the perturbation $\phi_1$. The same perturbation scheme can be extended to segments without any change. $DROP$ and $PSim$ are then calculated for each image and for the whole dataset as described in Equation (7) and Equation (6) respectively.

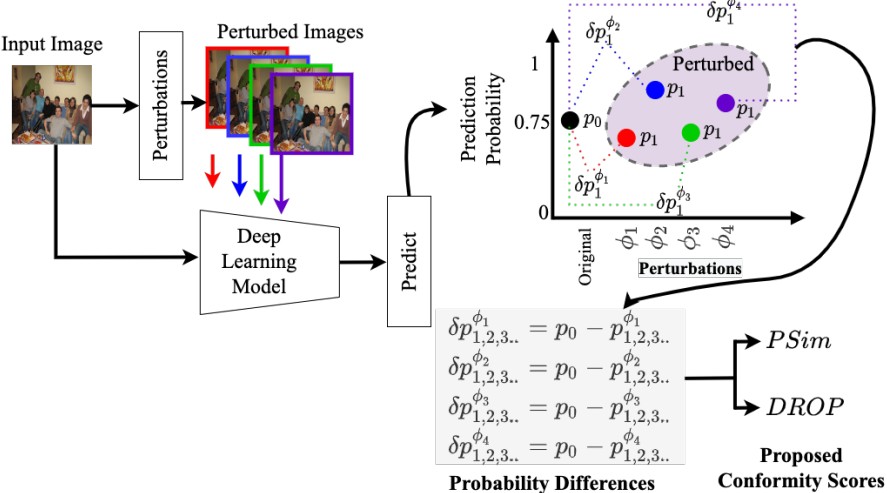

Figure 1: Proposed approach for estimating conformity scores of the deep learning models using the prediction probabilities on perturbed images.

---

**Algorithm 1** Algorithm for calculating $DROP$ and $PSim$

$p_0 \leftarrow model.predict(I)$       ▷ Unperturbed image $I$
$\{i\} \leftarrow S$       ▷ $i$ pixel in $S$ pixels
$\phi \leftarrow \{\Phi\}$       ▷ set of all perturbation types
$\mathcal{L} \leftarrow []$
$\mathcal{L}$       ▷ List of pixel importance ranks from all perturbation types
$\delta\mathcal{P} \leftarrow []$       ▷ $\delta\mathcal{P}$ is the $DROP$ score
**for all** $\phi$ **do**
     $\delta P \leftarrow []$
     **for all** $i$ $in$ $\{S\}$ **do**
        $I_i^{\phi} \leftarrow perturb\_image(I_i, \phi)$       ▷ for $i^{th}$ pixel in image $I$
        $p_i^{\phi} \leftarrow model.predict(I_i^{\phi})$
        $\delta p_i^{\phi} = p_0 - p_i^{\phi}$
        $\delta P.append(\delta p_i^{\phi})$
     **end for**
     $\delta\mathcal{P}.append(|\{\delta P \geq 0\}|)$       ▷ Append count of $\delta P \geq 0$
     $l \leftarrow argsort(\delta P)$
     $\mathcal{L}.append(l)$
**end for**
$rbo\_score \leftarrow pairwise\_rbo(\mathcal{L})$
**return** $\mu(\delta\mathcal{P})$, $\mu(rbo\_score)$       ▷ DROP (Equation (7)) and $PSim$ (Equation (6)) scores

---

## 4 EXPERIMENTAL SETUP

We use three pre-trained, and two adversarially trained image classification models, and three well-known datasets in our experiments. We conduct our analysis on InceptionV3 (Szegedy et al., 2016), Xception (Chollet, 2017), and ResNet50 (He et al., 2016) initialized with ImageNet weights. For, adversarial models we used the weights of adversarially trained ResNet50 architecture viz., ImageNet L2-norm (ResNet50) with $\epsilon = 3$ and ImageNet Linf-norm (ResNet50) with $\epsilon = 8/255$ ( refer Engstrom et al. (2019) for details). Imagenette from tensorflow.org (et.al.), Oxford-IIIT Pet Dataset (Parkhi et al., 2012) and PASCAL VOC 2007 (Everingham et al.) are used to conduct our experiments. The Imagenette dataset is a subset of the Imagenet (et.al.) dataset with ten easily classified classes. We used the validation part of this dataset for our experiments, which has around 3925 images. The Oxford-IIIT Pet Dataset (Parkhi et al., 2012) and PASCAL VOC 2007 (Everingham et al.) datasets did not have train and test splits. Hence, we considered all the images for

these two datasets, i.e., 7390 of the Oxford-IIIT Pet dataset and 4952 of the PASCAL VOC 2007 dataset. For each model, $predict$ was called for $(3925 + 7930 + 4952)$ $images \times 50$ $pixels \times 9$ $perturbatiotypes \times 2$ $perturbationschemes$ values, approximately, 15 million times, and in total, predict was called approximately 75 million times. Further, our goal was not to be exhaustive with different datasets and models but to understand the impact of perturbations to evaluate the fidelity of saliency maps from the perspective of PIR. Our code was written in Python 3.10 and Tensorflow 2.9 and for computing we leveraged A100 GPUs.

### 4.1 Perturbation Details

We considered nine different perturbation types i.e., two inpainting based perturbations for all our experiments. Specifically, we used Telea (Telea, 2004) and Navier Strokes (Bertalmio et al., 2001)), Gaussian Blur (three different widths of the Gaussian Kernel) and setting a random value, min, max and mean of the image pixel values as pixel values. The perturbations are represented as 'IT' (Telea inpainting),'IN' (Navier Strokes inpainting), 'FR' (setting pixel value randomly), 'U0' (image min), 'U1' (image max), 'U0.5' (image mean), 'G3' (Gaussian blur with kernel widths of 0.3), 'G9' (Gaussian blur with kernel widths of 0.9) and 'G1.5' (Gaussian blur with kernel widths of 1.5). Further, we perturb the pixels/segments using two perturbation schemes viz., pixel-wise and segment-wise. We use the property that a subset of a ranked order list maintains ranking and select 50 random pixels (refer to proof in Appendix S2). The same argument can be extended to segments as shown in our analysis.

## 5 Results and Discussion

### 5.1 DROP and PSim scores for all Perturbations

Table 1 shows the $DROP$ and $PSim$ values for different models over different datasets for pixel-wise perturbation scheme. The chosen models, i.e., Inception V3, Xception, and ResNet50 pre-trained with Imagenet weights. As seen in Table 1, it can be observed that the $DROP$ values are around 0.5 to 0.6 for all models across datasets. This indicates that only for 50 % to 60% of the pixels, the probability dropped on perturbation. This invalidates Point [P1] of the assumption in Section 2. Further, Table 1 shows the $PSim$ values for all the models over all datasets. As seen from the table, the $PSim$ values are small, but as per Equation (9), they should have been $\approx 1$. This invalidates Point [P2] of the assumption in Section 2. Further, this observation is consistent for all three models and across all datasets for segment-wise perturbation scheme as seen in Table 1. Thus, for different perturbations, the mentioned models will not conform to the assumptions made by the perturbation based fidelity metrics.

Further, we show the $DROP$ and $PSim$ scores for the adversarially trained ResNet50 models for both perturbation schemes in Table 2. Both $DROP$ and $PSim$ scores are much lower than 1 in all cases, and hence, adversarial training does not necessarily result in consistency of fidelity metrics. Due to the unavailability of adversarially trained models for Inception_V3 and Xception architectures, we had to limit our experiments to ResNet50 architecture. Hence, we refrain from making conclusive remarks regarding the consistency of fidelity metrics with respect to adversarially trained models.

### 5.2 DROP for Individual Perturbations

We present the distribution of $DROP$ scores for Inception V3, Resnet50, and Xception models in the Imagenette dataset in Figure 2. For all perturbations, except the variants of Gaussian Blur, the $DROP$ scores have the highest density at around 0.5. However, the variations of the Gaussian Blur for the ResNet50 model seem to be closer to 1. This pattern is similar for other datasets (Figure S2 and Figure S3 in supplementary). Further, we estimated the probability of the $DROP$ scores to be closer to 1 (i.e., above the cut-offs of 0.80, 0.85, 0.90, and 0.95) by using Kernel Density Estimation (KDE), with Scott's rule Scott (2015) for bandwidth calculation, owing to its non-parametric nature. In Figure 3, we show the estimated probabilities for $DROP$ and $PSim$ scores across all datasets, models, and perturbation types for segment-wise perturbation to be $\geq 0.8$. The first two letters of model name and dataset name are used along the axis for "Dataset - Model" to represent their

Table 1: $DROP$ and $PSim$ scores across all datasets, models, perturbations for pixel-wise perturbation scheme and segment-wise perturbation scheme. The segments were computed using the Quickshift (Vedaldi & Soatto, 2008) segmentation algorithm. The results are shown as Mean $\pm$ Standard Deviation. Ideal value $DROP$ and $PSim$ should be 1 and higher the better.

| Dataset | | Inception | Xception | ResNet |
|---------|---|-----------|----------|--------|
| | | Pixel-wise perturbation | | |
| Imagenette | $DROP$ | 0.504±0.131 | 0.514±0.134 | 0.643±0.153 |
| | $PSim$ | 0.432±0.181 | 0.431±0.185 | 0.570±0.298 |
| Oxford Pets | $DROP$ | 0.507±0.130 | 0.504±0.138 | 0.636±0.132 |
| | $PSim$ | 0.428±0.183 | 0.430±0.186 | 0.582±0.289 |
| VOC2007 | $DROP$ | 0.511±0.115 | 0.550±0.180 | 0.512±0.132 |
| | $PSim$ | 0.643±0.130 | 0.433±0.189 | 0.573±0.301 |
| | | Segment-wise perturbation | | |
| Imagenette | $DROP$ | 0.515±0.135 | 0.518±0.126 | 0.553±0.111 |
| | $PSim$ | 0.310±0.181 | 0.269±0.142 | 0.329±0.179 |
| Oxford Pets | $DROP$ | 0.507±0.120 | 0.516±0.095 | 0.546±0.107 |
| | $PSim$ | 0.255±0.129 | 0.307±0.179 | 0.309±0.181 |
| VOC2007 | $DROP$ | 0.542±0.102 | 0.517±0.091 | 0.529±0.100 |
| | $PSim$ | 0.267±0.166 | 0.294±0.179 | 0.299±0.182 |

Table 2: $DROP$ and $PSim$ scores for adversarially trained ResNet50 models (Linf-norm and L2-norm) for pixel-wise and segment-wise perturbation schemes. (*Higher scores are better)

| | Pixel-wise Perturbation Scheme | | | |
|---------|--------|----------|--------|----------|
| Dataset | L2-norm | Linf-norm | L2-norm | Linf-norm |
| Imagenette | 0.555±0.374 | 0.555±0.357 | 0.237±0.140 | 0.209±0.097 |
| Oxford Pets | 0.580±0.369 | 0.567±0.369 | 0.217±0.133 | 0.186±0.116 |
| VOC2007 | 0.528±0.383 | 0.546±0.371 | 0.243±0.124 | 0.181±0.106 |
| | Segment-wise Perturbation Scheme | | | |
| Imagenette | 0.574±0.238 | 0.526±0.220 | 0.321±0.173 | 0.301±0.146 |
| Oxford Pets | 0.541±0.218 | 0.567±0.213 | 0.318±0.165 | 0.326±0.182 |
| VOC2007 | 0.557±0.186 | 0.517±0.181 | 0.292±0.148 | 0.289±0.155 |

combinations. In most scenarios, the estimated probabilities for $DROP$ are low, but the variants of Gaussian Blur show relatively higher probabilities than other perturbations. We see a similar trend for the segment-wise perturbation scheme (refer Figure S14 in supplementary) and for the different cut-offs of estimate probabilities in Figure S12, Figure S13 of supplementary. This demonstrates empirically that fidelity metrics have low conformity to Point [P1].

### 5.3 PSim for Individual Pairs of Perturbations

The pairwise $PSim$ scores for all perturbation pairs corresponding to the Inception V3 model on the Imagenette dataset are shown for the pixel-wise perturbation scheme in Figure 4. Most of the perturbation pairs have low $PSim$ scores, but for the three pairs of Gaussian Blur (i.e., G3_G9, G3_G15, and G9_G15) and the pair for inpainting (IT vs. IN), the $PSim$ scores are relatively higher. We show the $PSim$ scores for all perturbation pairs on all dataset:model combinations in supplementary (Figure S4 - Figure S11). Further, the same trend is visible when we estimate the probability of $PSim$ scores to be $\geq 0.8$ (like Section 5.2). We show the surface plot of the estimated probabilities of $PSim$ scores to be $\geq 0.8$ for all perturbation pairs in Figure 5. The first two letters of model name and dataset name are used along the axis for "Dataset - Model" to represent their combinations. The results in Figure 5 are similar to the observations of Figure 4. However, it has to be noted that in none of the scenarios, $PSim$ score is $\approx 1$, indicating low conformity to Point [P1]. We see a similar trend for the segment-wise perturbation scheme (refer Figure S15 in supplementary). Hence, the ranks of the pixels/segments (as mentioned in Section 2.1) would vary for different perturbation types and lead to inconsistency in fidelity metrics.

From the low probabilities observed in Section 5.2, and Section 5.3, it can be established that fidelity metrics have low conformity to Point [P1] and hence are not consistent across a wide variety of perturbations. As such, it is imperative to specify the perturbation type to be used when reporting the fidelity scores from these fidelity metrics. The perturbation type can be determined using domain-

related theoretical reasoning and/or empirically (as discussed in (Bora et al., 2024)). Further, we also observed that, out of the perturbation types considered, Gaussian Blur was relatively consistent compared to other perturbation types as it had higher scores for both conformity measures.

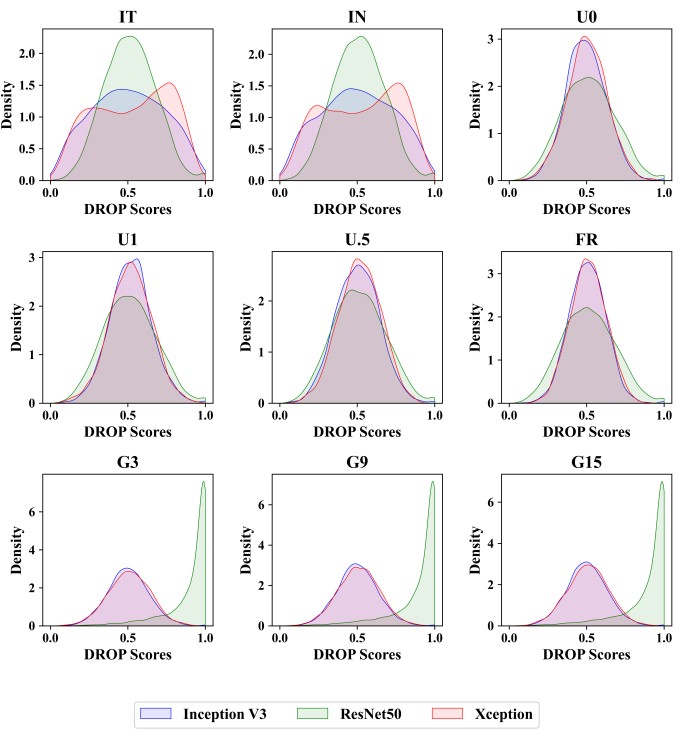

Figure 2: Distribution of $DROP$ scores across all models, perturbation types using pixel-wise perturbation scheme for Imagenette Dataset

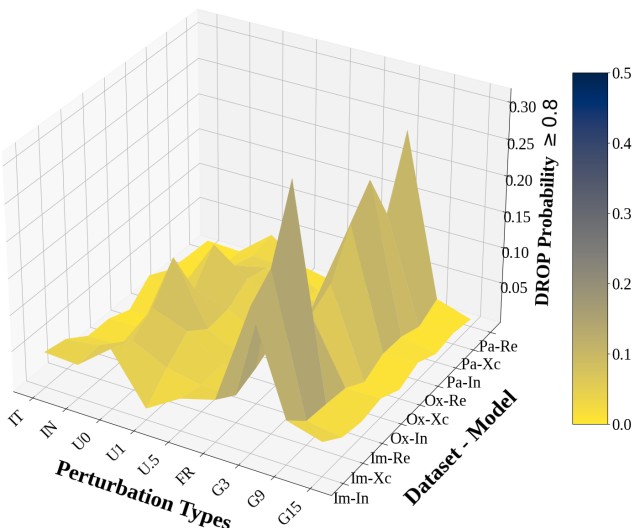

Figure 3: Surface Plot of $DROP$ scores' probabilities to be above 0.8 for all datasets, models, and perturbation types using pixel-wise perturbation scheme

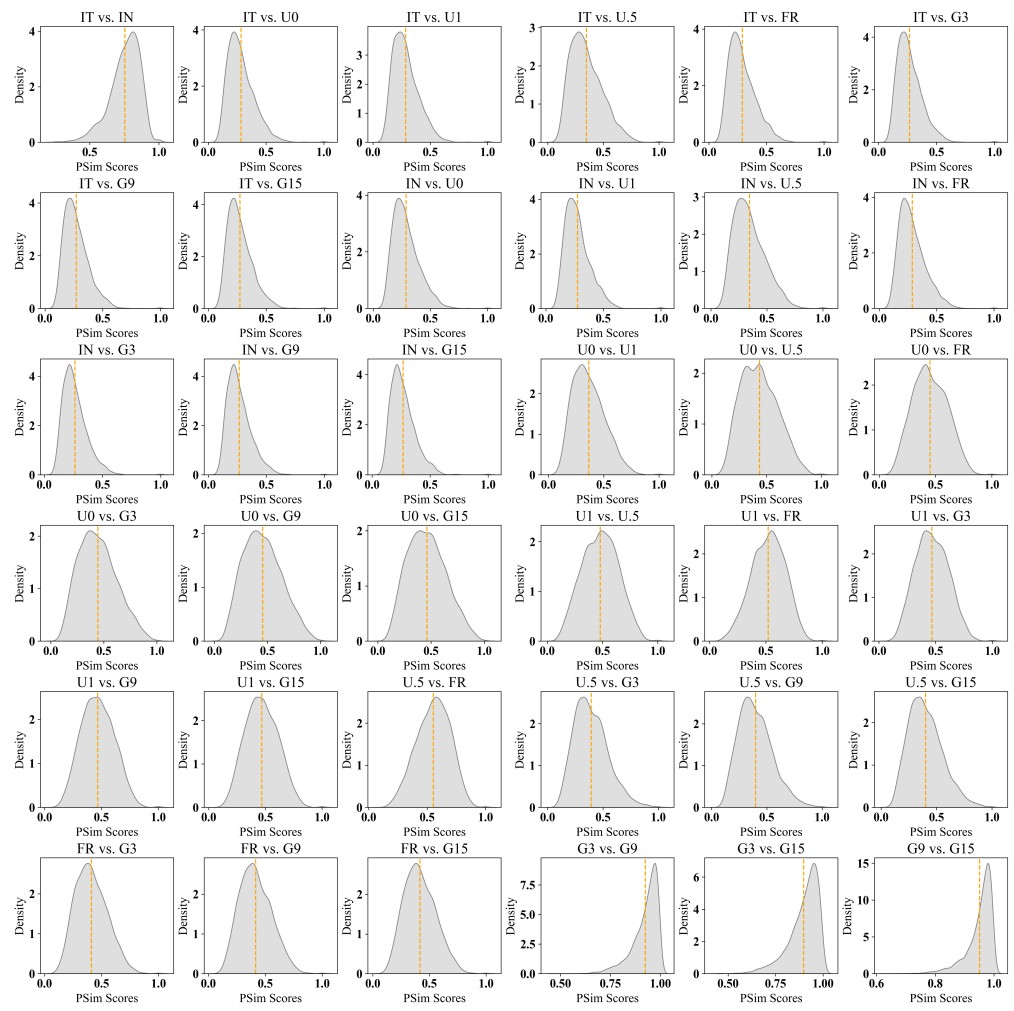

Figure 4: Distribution of pairwise $PSim$ scores for all perturbation types for Inception V3 model using pixel-wise perturbation scheme on Imagenette Dataset

## 6 CONCLUSION AND FUTURE WORK

The prediction probability of DL models varied significantly for the same image and the same model for the considered perturbations. This variation in the output probabilities led to a high variance in the PIR. Thus, the metrics that implicitly rely on the invariance of PIR for measuring fidelity would be rendered unreliable and fail the sanity checks. While previous studies have limited the analysis of unreliability to the metric level, we demonstrated that unreliability arises as a property of the DL models with respect to perturbations. Thus, we recommend using the proposed metrics as a preconditional check before analyzing the fidelity of saliency maps. Further, we advocate specifying the perturbation type while reporting fidelity scores from these fidelity metrics. However, out of the considered perturbation, Gaussian Blur was relatively consistent compared to other perturbation types. Future works should consider the high variance in PIR and the lack of robustness around the predicted instance to devise reliable fidelity metrics. In the future, we plan to extend our study to analyze the behavior of adversarially trained DL models concerning perturbations for different architectures using the proposed conformity measures.

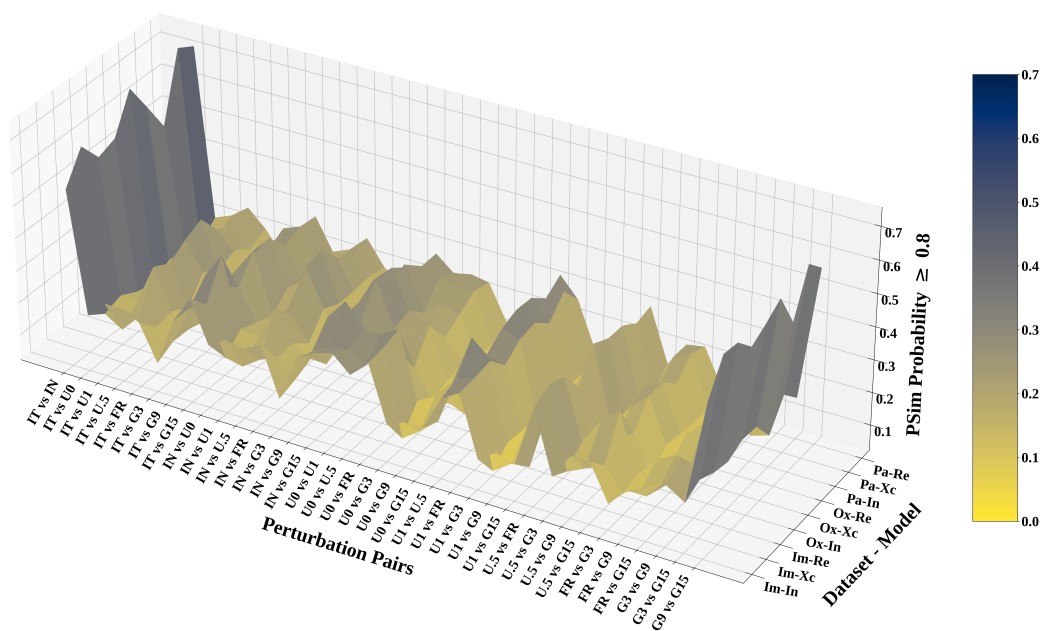

Figure 5: Surface Plot of $PSim$ scores' probabilities to be above 0.8 for all datasets, models, and perturbation types using pixel-wise perturbation scheme

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
