# OpenReview forum: "Examining Why Perturbation-Based Fidelity Metrics are Inconsistent"
_ICLR.cc/2025/Conference — Submitted to ICLR 2025_

### Official Review · Reviewer_dEgy · 2024-10-31

**Soundness:** 2
**Presentation:** 2
**Contribution:** 2
**Rating:** 3
**Confidence:** 5

**Summary:**

This paper examines the inconsistency of perturbation-based fidelity metrics for quantifying saliency-based XAI modes.

**Strengths:**

1. This paper explores an interesting and important ML problem: quantitatively measuring the consistency of fidelity metrics in the presence of various perturbations.

**Weaknesses:**

1. The paper presents limited technical novelty. Primarily, it is more essential to establish the accuracy of saliency-based XAI models, which this work overlooks. Kindly refer to the FunnyBirds dataset (https://arxiv.org/abs/2308.06248) for more context.

2. Building on Comment 1, is it necessary to assert [P1] and [P2] if the saliency map does not accurately reflect the decision rule learned by the classifier?

3. The two proposed measures are already widely used in related fields.

4. Mathematical notations are not consistently applied. For instance, in Eq. (2), the subscripts in $p$ (i.e., $0$ and $i$) have different meanings. Additionally, the meaning of $\rho$ in $R_\rho$ on Line 148 is unclear.

5. After reviewing the entire manuscript, the reasons behind the inconsistency of the perturbation-based fidelity metrics remain unclear.

6. Are there any principled approaches to select perturbations when assessing the consistency of fidelity metrics?

**Questions:**

1. The authors may clearly discuss prediction consistency in the context of prediction correctness. What if the prediction is not correct, should we still insist on prediction consistency?
2. The novelty of the two proposed measures should be better stated.
3. The reasons behind the inconsistency of the perturbation-based fidelity metrics should be better itemized.

---

### Official Review · Reviewer_YDNe · 2024-11-01

**Soundness:** 3
**Presentation:** 2
**Contribution:** 3
**Rating:** 6
**Confidence:** 2

**Summary:**

The study investigates fidelity metrics, a method to validate saliency maps, which are visual explanations of model decisions. These metrics traditionally rely on perturbation techniques that rank pixel importance by assessing the change in model output when pixels are modified. However, prior studies noted inconsistencies in these metrics across perturbations without explaining their origins. This paper addresses the root causes of these inconsistencies by analyzing different fidelity metrics and the underlying assumptions they rely on.

**Strengths:**

1. Detailed Analysis: The study is comprehensive, covering multiple perturbation techniques and datasets to provide robust findings.
2. Proposed Measures: DROP and PSim offer a standardized way to examine fidelity metric consistency, a valuable tool for the XAI community.
3. Clear Recommendations: The authors make actionable recommendations, like specifying the perturbation type when using fidelity metrics, to improve reliability.

**Weaknesses:**

**1. Limited Analysis on Robustness of Proposed Measures (DROP and PSim)**

While DROP and PSim are proposed to assess fidelity metric consistency, the paper does not fully explore the sensitivity of these measures themselves to various perturbations and model types. Without an in-depth robustness analysis, it is unclear if DROP and PSim are reliable across a broader set of tasks and perturbation techniques. I suggest authors clarify this point.

**2. Narrow Scope of Adversarial Analysis**

The paper only uses adversarially trained models within the ResNet50 architecture, due to unavailability for other architectures. This limited scope restricts the generalizability of findings regarding the impact of adversarial training on fidelity metric consistency. Expanding to include other architectures, or a more diverse range of adversarial techniques, could provide stronger support for the conclusions. The authors are encouraged to add more experiments.

**3. Complexity of Theoretical Framework**

 The theoretical background section and mathematical formulations are quite dense, potentially limiting accessibility for readers unfamiliar with fidelity metrics or perturbation techniques. Simplifying explanations, using visualizations, or adding intuitive examples could improve comprehension and broaden the paper’s audience. Improving the presentation and reorganizing some important figures ( such as Figure 1) can help readers quickly grasp the ideas.

**Questions:**

1. Could the authors provide more details or experiments to examine the sensitivity of DROP and PSim under various conditions? Such an analysis would clarify whether these measures are reliable for a wider set of tasks and perturbation techniques.

2. The study focuses on adversarially trained models within the ResNet50 architecture due to availability constraints. This limited scope raises questions about the generalizability of the findings, especially regarding adversarial training's impact on fidelity metric consistency. Are there plans to test other architectures or adversarial techniques to strengthen the conclusions? If additional experiments are not feasible, could the authors discuss the implications of this limitation?

3. The theoretical background and mathematical formulations are dense, potentially challenging for readers less familiar with fidelity metrics or perturbation techniques. Could the authors consider simplifying these sections or adding intuitive examples and visualizations to improve accessibility? Additionally, reorganizing key figures, such as Figure 1, might help readers quickly understand the main concepts.

---

### Official Review · Reviewer_XzXP · 2024-11-04

**Soundness:** 1
**Presentation:** 2
**Contribution:** 2
**Rating:** 3
**Confidence:** 3

**Summary:**

The paper investigates inconsistencies in perturbation-based fidelity metrics, commonly used to evaluate the fidelity of saliency maps in explainable AI (XAI). The authors argue that existing fidelity metrics, such as Area Over the Perturbation Curve (AOPC) and Average Drop, may be unreliable due to their inherent assumptions, specifically in how they handle different types of perturbations. They introduce two new conformity measures, DROP and PSim, designed to assess the fidelity of saliency maps more consistently across different perturbation schemes. The study uses multiple perturbation types and models to demonstrate these metrics' effectiveness. Ultimately, the authors propose using their conformity measures as preconditions for saliency map fidelity evaluation and recommend specifying the perturbation type when reporting fidelity scores.

**Strengths:**

1. The paper provides an extensive empirical evaluation of fidelity metrics using nine perturbation types across multiple models and datasets. This wide-ranging approach strengthens the validity of the findings.
2. By introducing DROP and PSim, the authors offer a approach that highlights discrepancies in fidelity assessment, paving the way for more consistent interpretations in future XAI research.

**Weaknesses:**

1. (**Major**) There is a notable discrepancy between the paper's proposed metrics (DROP and PSim) and the operational principles of traditional fidelity metrics. The study uses measurements based on fixed performance drops at predetermined steps, while established fidelity metrics like AOPC and ROAR/ROAD accumulate effects across perturbation steps. This accumulation is essential because natural image pixels are often interdependent, particularly spatially. Existing fidelity metrics intentionally incorporate cumulative effects to account for these dependencies, which this paper’s approach overlooks. This gap suggests a misalignment between the criticism of traditional metrics and the design philosophy behind DROP and PSim.
2. (**Major**) The paper suggests that fidelity metrics should ideally exhibit consistency across different perturbations, but it lacks theoretical or empirical justification for this claim. It seems natural that selecting different perturbations would yield varied feature rankings, as other fidelity metrics typically fix a single perturbation setting to report performance. If the authors assert that a PSim score of 1 is desirable for fidelity metrics, they need to provide a basis for why this consistency is necessary.
3. (**Major**) The paper appears to overlook important ongoing discussions in the field regarding fidelity metrics. For instance, recent research has highlighted the need to differentiate between performance drops caused by feature perturbations and those stemming from out-of-distribution (OOD) effects ([Hooker, 2019], [Rong, 2023]). By not addressing this distinction, the paper's propositions seem to bypass a crucial aspect of the fidelity metric discourse.
4. (Minor) The segment-wise approach in the paper is insufficiently detailed, leaving unclear how this method was specifically implemented. This lack of transparency makes it challenging to replicate or evaluate the effectiveness of the segment-wise perturbation scheme as proposed.
5. (Minor) The visualizations in Figures 3 and 5 are problematic due to discontinuities along the axes. Given the nature of the data, using surface plots with disconnected axes seems unnecessary and hinders the clarity of the results. A different visualization method may be more effective in conveying the findings.

### References
- ROAR [Hooker, 2019] Hooker, Sara, et al. "A benchmark for interpretability methods in deep neural networks." Advances in neural information processing systems 32 (2019).
- ROAD [Rong, 2023] Rong, Yao, et al. "A consistent and efficient evaluation strategy for attribution methods." In ICML. 2023.

**Questions:**

- How does the fidelity metric’s conformance to PSim, specifically aiming for a score close to 1, contribute to reliability? Is there a theoretical basis that justifies PSim’s validity as a fidelity metric?
- Why was there no mention of existing literature that addresses distinctions between fidelity metrics and OOD-related performance drops? How might these distinctions impact the interpretation of the proposed conformity measures?
- Could you clarify the specific methodology behind the segment-wise perturbation scheme? This detail would enhance the replicability and understanding of the proposed approach.

---

### Official Review · Reviewer_5vt1 · 2024-11-04

**Soundness:** 2
**Presentation:** 1
**Contribution:** 2
**Rating:** 3
**Confidence:** 4

**Summary:**

The paper proposes two metrics to evaluate the consistency of perturbation-based fidelity metrics. Specifically, the two metrics are: Drop in Prediction Probability (DROP) and Pixel Rank Similarity (PSim). These metrics are evaluated on numerous deep learning (DL) models using many perturbation types.

While there are some interesting ideas, the present form of the paper does not justify a higher score.

**Strengths:**

1. The premise of the problem is interesting.

2. The proposed conformity metrics are simple.

3. Performance evaluation on CNNs shows that they are inconsistent with respect to perturbations as measured using the proposed DROP and PSim metrics.

**Weaknesses:**

1. The practical utility of the proposed metric is not clear. While it is recommended that DROP and PSim be measured prior to analyzing the fidelity of saliency maps, there is no guidance on how these measures could be used to interpret saliency maps. Such guidance would have been very helpful and have added to the usefulness of the proposed metrics.

2. The quality of writing needs significant improvement. Specifically, the definitions and metrics introduced in Sections 2.1 to 2.3 need a careful proofread to find and fix notational inconsistencies and errors. For example, (9) does not have the summation index on the right-hand side. The authors are encouraged to learn about and follow standard conventions used for equation writing.

3. Please proofread the paper for typographical errors. For example, “varaiances” -> “variances”.

4. The behavior of Gaussian blur could be explained better.

5. The choice of DL models used for this study has not been justified. Also, transformer models have not been considered at all.

**Questions:**

1. Can you please provide visual examples to illustrate the utility of the proposed metrics? This could include the original image, the perturbed image, and a saliency map that can be analyzed/explained using the DROP and PSim scores.

2. Why have transformer models not been considered given their widespread use in computer vision models? For example, open-source ViT implementations are readily available. This would add further value to the work.

3. Can you please provide guidance on how the DROP and PSim scores should be interpreted and used when working with saliency maps?

4. Can you please label DROP and PSim scores in Table 2?

---

### Meta-Review · Area_Chair_T58z · 2024-12-18

**Metareview:**

In this paper, the authors presented a study about the inconsistencies in the perturbation-based fidelity metrics. Specifically, two measures were proposed to examine existing fidelity metrics across different datasets. The analysis showed that the studied fidelity metrics are inconsistent and unreliable.
The strengths of this paper include:
- The problem investigated is interesting and potentially important in ML.
- The proposed conformity measures are simple and offer a way to examine consistency in fidelity metrics
- Using the proposed measures, a detailed analysis was presented with recommendations.

The weaknesses of this paper are:
- The proposed measures/metrics were not clearly presented nor extensively investigated. The practical utility, sensitivity to different perturbations and model types, and insufficient analysis are the main concerns from the reviewers.
- The discrepancy between the proposed measures and previous fidelity metrics. The misalignment led to potential issues when compared to traditional metrics.
- Some key related works were missing in the discussion in this paper (details please see the review comments).
- The analysis is a bit limited. Only models trained using ResNet50 architecture were considered, and the choice of the models was not clearly justified, limiting the findings presented.
- The writing of the paper also needs a major revision.

The paper received 3 Reject and 1 borderline Accept.
While there are some interesting points introduced by this paper, considering the major concerns raised by the reviewers and the missing rebuttal from the authors, concerns were not cleared, and the paper in its current form is not ready to be presented at ICLR.

**Additional Comments On Reviewer Discussion:**

There was no rebuttal provided by the authors nor discussions between the authors and reviewers. The major concerns raised by the reviewers were not cleared as a result. The final decision was made based on the remained major concerns and the above justification.

---

### Decision · Program_Chairs · 2025-01-22

Reject